

# Charges in the extended BMS algebra: Definitions and applications

**Massimo Porrati**

Center for Cosmology and Particle Physics, Department of Physics, New York University,
726 Broadway, New York NY 10003, USA

*4th International Conference on Holography,
String Theory and Discrete Approach
Hanoi, Vietnam, 2020*

## Abstract

This is a review of selected topics from recent work on symmetry charges in asymptotically flat spacetime done by the author in collaboration with U. Kol and R. Javadinezhad. First we reinterpret the reality constraint on the boundary graviton as the gauge fixing of a new local symmetry, called dual supertranslations. This symmetry extends the BMS group and bears many similarities to the dual (magnetic) gauge symmetry of electrodynamics. We use this new gauge symmetry to propose a new description of the TAUB-NUT space that does not contain closed time-like curves. Next we summarize progress towards the definition of Lorentz and super-Lorentz charges that commute with supertranslations and with the soft graviton mode.

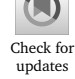

## 1 Introduction

The BMS group of symmetries of asymptotically flat spacetime dates from the 1960's [1,2] but its extension to include *dual supertranslations* was discovered only much later [3], while the action of supertranslations on phase space [4] or their interpretation as gauge symmetries [5] were studied only in the last two year. With the benefit of a healthy dose of hindsight it is surprising that it took so long to discover all that, since gauged supertranslations arise naturally from studying the same constraints on asymptotic dynamics that define the BMS group, as we will see now.

The expansion in inverse powers of the radial coordinate $r$ of an asymptotically flat metric

is

$$
\begin{aligned}
ds^2 &= -du^2 - 2du\, dr + r^2\left(h_{AB} + \frac{C_{AB}}{r}\right)d\Theta^A d\Theta^B + D^A C_{AB}\, du\, d\Theta^B + \frac{2m_B}{r}du^2 + D^B C_{AB} du d\Theta^A \\
&\quad + \frac{1}{16r^2}C_{AB}C^{AB} du dr + \frac{1}{r}\left(\frac{4}{3}(N_A + u\partial_A m_B) - \frac{1}{8}\partial_A\left(C_{BD}C^{BD}\right)\right)du d\Theta^A \\
&\quad + \frac{1}{4}h_{AB}C_{CD}C^{CD}d\Theta^A d\Theta^B + \ldots,
\end{aligned}
\tag{1}
$$

where $C_{AB}$ is symmetric and traceless while the dots in (1) denote subdominant terms in $1/r$. The Poisson brackets derived from the Einstein action [6–9] show that an appropriate choice of dynamical variables is given by the Bondi news $N_{AB} = \partial_u C_{AB}$ and the boundary graviton $C_{AB}^\infty$. The Poisson brackets of the Bondi news are

$$
\{N_{AB}(u,\Theta), N^{CD}(u',\Theta')\} = 16\pi G\, \delta_{AB}^{CD}\partial_u \delta(u-u')\delta^2(\Theta-\Theta'),
\tag{2}
$$

where $\delta_{AB}^{CD} \equiv \delta_A^C \delta_B^D + \delta_A^D \delta_B^C - h_{AB}h^{CD}$. The Poisson brackets of the boundary graviton depend on its definition. The supertranslation charge is [10–12]

$$
\begin{aligned}
Q[f] &= Q_h[f] + Q_s[f], \\
Q_h[f] &= \frac{1}{4\pi}\int_{\mathcal{I}^+} du\, d^2\Theta \sqrt{h}\, f(\Theta)T_{uu}, \qquad Q_s[f] = -\frac{1}{16\pi G}\int_{\mathcal{I}^+} du\, d^2\Theta \sqrt{h}\, f(\Theta)D^A D^B N_{AB}, \\
T_{uu} &= \frac{1}{8G}N_{AB}N^{AB} + \lim_{r\to\infty} r^2 T_{uu}^M, \qquad T^M = \text{matter stress-energy tensor.}
\end{aligned}
\tag{3}
$$

If both components of $C_{AB}$ were independent and all Poisson brackets were continuous in the limit $u \to -\infty$ the supertranslation charge would not generate coordinate transformations on the shears $C_{AB}$: $\{Q, C_{AB}\} \neq \mathcal{L}_f C_{AB}$ ($\mathcal{L}_f$ = Lie derivative along the vector $f$). To solve this problem ref. [13] proposed to restrict the boundary graviton to be pure gauge

$$
C_{AB}^\infty = (D_A D_B + D_B D_A - h_{AB}D^2)C.
\tag{4}
$$

In complex coordinates, $z, \bar{z}$, the two independent components of the shear are $C_{zz}$ and $C_{\bar{z}\bar{z}} = \overline{(C_{zz})}$ so that equation (4) becomes

$$
C_{zz}^\infty = D_z D_z C, \qquad C_{\bar{z}\bar{z}}^\infty = D_{\bar{z}}D_{\bar{z}}C, \qquad \text{Im}\, C = 0.
\tag{5}
$$

The most general boundary graviton is parametrized by a *complex* scalar $C$. It is natural to think of the condition $\text{Im}\, C = 0$ as a gauge fixing of the symmetry

$$
C(\Theta) \to C(\Theta) + i f(\Theta), \qquad f(\Theta) \in \mathbb{R}.
\tag{6}
$$

The generator of this symmetry is the dual supertranslation charge [3–5]

$$
M(f) = \frac{i}{16\pi G}\int d^2 z f(z,\bar{z})(D_{\bar{z}}D^z C_{zz} - D_z D^{\bar{z}}C_{\bar{z}\bar{z}})\bigg|_{u=-\infty}.
\tag{7}
$$

So the dual supertranslation charge (7) arises naturally from relaxing the boundary condition $\text{Im}\, C = 0$. It is always possible to introduce a gauge symmetry by adding gauge degrees of freedom that can be removed by a gauge transformation, so the charge $M(f)$ certainly exists. But if it is a gauge charge then it is either zero or constant on any irreducible representation of the algebra of observables. The question that we will address next is if anything is gained by introducing a gauge symmetry and removing it by the gauge fixing $\text{Im}\, C = 0$.

## 2 Taming Taub-NUT

An analogy with the Abelian Higgs model may help here. The degrees of freedom are a complex scalar $\phi$ and an Abelian gauge field $A_\mu$. The scalar potential $U(\phi) = \lambda(|\phi|^2 - v^2)^2$ has minima at $|\phi| = v$ that break the gauge symmetry. Whenever $\phi \neq 0$ a good gauge fixing is given by the condition $\text{Im}\,\phi = 0$ but not all regular solutions of the classical equations of motion obey $\phi \neq 0$. A famous example is the Abrikosov-Nielsen-Olesen (ANO) string [14, 15]. It describes a string extending along an infinite straight line. It is regular everywhere when the fields have the following behavior at large and small radius (in cylindrical coordinates $0 \leq r < +\infty$, $\theta \sim \theta + 2\pi$, $-\infty < z < +\infty$)

$$\lim_{r \to +\infty} A_\theta = n, \qquad \lim_{r \to +\infty} \phi = v e^{in\theta}, \qquad \lim_{r \to 0} A_\theta = 0, \qquad \lim_{r \to 0} \phi = 0. \tag{8}$$

When we transform the ANO string to the gauge $\text{Im}\,\phi = 0$ we create an unphysical singularity at $r = 0$, so in this case $\text{Im}\,\phi = 0$ is not a globally well-defined gauge. This analogy helps us to get a new prospective on the Taub-NUT solution of Einstein's equations.

The Taub-NUT metric is

$$ds^2 = -f(r)(dt + 2l\cos\theta\, d\varphi)^2 + \frac{dr^2}{f(r)} + (r^2 + l^2)(d\theta^2 + \sin^2\theta\, d\varphi^2), \tag{9}$$

where $f(r) = (r^2 - 2mr - l^2)/(r^2 + l^2)$. Here $m$ is the mass aspect and $l$ is called the NUT parameter. The Taub-NUT metric has two horizons located at $r_\pm = m \pm \sqrt{m^2 + \ell^2}$. The Taub region is at $r_- < r < r_+$ while the NUT regions are the domains $r > r_+$ and $r < r_-$.

The Taub-NUT metric contains a string-like singularity along the axes $\theta = 0$ and $\theta = \pi$. It is possible to remove the singularity in the "North" hemisphere $0 \leq \theta \leq \pi/2$ using the change of coordinates $t \to t - 2\ell\varphi$, which casts the metric into the form

$$ds_N^2 = -f(r)\left(dt_N - 4l\sin^2\frac{\theta}{2}d\varphi\right)^2 + \frac{dr^2}{f(r)} + (r^2 + l^2)(d\theta^2 + \sin^2\theta\, d\varphi^2). \tag{10}$$

We can similarly remove the singularity in the South hemisphere $\pi/2 \leq \theta \leq \pi$ with the change of coordinates $t \to t + 2\ell\varphi$, resulting in the metric

$$ds_S^2 = -f(r)\left(dt_S + 4l\cos^2\frac{\theta}{2}d\varphi\right)^2 + \frac{dr^2}{f(r)} + (r^2 + l^2)(d\theta^2 + \sin^2\theta\, d\varphi^2). \tag{11}$$

A globally regular solution is obtained by identifying the two metrics along the equator $\varphi = \pi/2$ up to a diffeomorphism (which is a gauge transformation)

$$t_N = t_S + 4l\varphi. \tag{12}$$

Since $\varphi$ is compact with a period of $2\pi$ then both $t_N$ and $t_S$ have to be compact with a period $8\pi l$ so the solution contains closed timelike curves (CTSs).

Promoting the symmetry (6) to a gauge symmetry we obtain an alternative construction of an everywhere-regular solution (asymptotically) free of the CTCs curves due to eq. (12): instead of identifying the metric up to a spacetime diffeomorphism we identify it up to a dual supertranslation symmetry $C_N = C_S + if$. For the metrics in (10,11) we find [5]

$$C_N = 8il\log\cos\frac{\theta}{2}, \qquad C_N = 8il\log\cos\frac{\theta}{2}, \qquad f = -8l\log\tan\frac{\theta}{2}. \tag{13}$$

A few remarks are necessary here.

- The gauge symmetry we proposed is a *conjecture*. For Taub-NUT all components of the expansion of the dual supertranslation charge in spherical harmonics are zero, except $l = 0, 1$. This is good, but not good enough, because *all* components of a gauge charge must act trivially on observables. Yet at face value there exist observables in general relativity that transform nontrivially under (6) –an obvious example is the metric itself. In [5] we showed that the dual supertranslation charge (7) acts trivially on the S-matrix and that the equations of motion of point particles in the limit $r \to +\infty$ are invariant under (6).

- A method to make dual supertranslations act trivially on the dynamical variables of general relativity is to *define* the boundary graviton as

$$C_{zz}^\infty = \frac{A}{2} \lim_{u \to -\infty} \left[ C_{zz}(u) + D_z^2 D_{\bar{z}}^{-2} P C_{\bar{z}\bar{z}}(u) \right] + \frac{A}{2} \lim_{u \to +\infty} \left[ C_{zz}(u) + D_z^2 D_{\bar{z}}^{-2} P C_{\bar{z}\bar{z}}(u) \right]. \quad (14)$$

  Here $P = 1 - Q$, with $Q$ the projection on the kernel of $D_{\bar{z}}^2$ while the proportionality constant $A$ is fixed to $A = 1$ by requiring continuity of the Poisson brackets $\{C_{zz}(u), C_{\bar{z}\bar{z}}\}$ in the limit $u \to \pm\infty$.

- It is an important open problem to define the dual supertranslation gauge symmetry everywhere in spacetime instead of giving a definition valid only asymptotically, as we have done here. Needless to say, this step is necessary to prove that CTCs *and singularities* are truly absent from Taub-NUT.

## 3   New Lorentz charges and open questions

The Lorentz charges do not commute with supertranslations. The $l = 0, 1$ harmonics of $Q[f]$ in eq. (3) are standard translations, so they are not expected to commute with the generators of the Lorentz algebra anyway. Supertranslations commute among themselves so they commute with spacetime translations. Therefore, after quantization vacuum states are degenerate and are $L^2$ function $\Psi$ with harmonics $\{C_{lm}|l > 1, -l \le m \le l\}$. The $l > 2$ harmonics shift the boundary graviton because of the commutation relation $\delta C_{lm} = \sum_{L>1,-L\le M\le L} f_{LM}\{Q_{LM}, C_{lm}\} = f_{lm}$ so a supertranslation generically transforms a vacuum, say an eigenstate of $C_{lm}$, into a different vacuum state. Because Lorentz transformations do not commute with supertranslations we find that therefore the definition of Lorentz charges, including angular momentum $\vec{J}$ is ambiguous. This can be seen clearly by considering a vacuum with zero angular momentum, $\Psi_0$. By definition $\vec{J}\Psi_0 = 0$ but since $[\vec{J}, Q_{lm}] \ne 0$ for $l > 1$, we also have other vacuum states $\Psi = (1 + \sum_{lm} f_{lm}Q_{lm})\Psi_0$. Each one of them is of course as good a vacuum as $\Psi_0$ but on them, generically, $\vec{J}(1+\sum_{lm} f_{lm}Q_{lm})\Psi_0 = \sum_{lm} f_{lm}[\vec{J}, Q_{lm}]\Psi_0 \ne 0$. So, even the apparently innocuous question: "what is the angular momentum of the vacuum?" seems to have no answer. Another route to discover that the angular momentum is not unambiguously defined in general relativity can be found in [17].

The next obvious question is whether a "better" definition of Lorentz charges exists. By better we mean a definition that commutes with the $l > 1$ harmonics of supertranslations. In this section we will show that for angular momentum such a definition exists and we will give an explicit construction of such a charge. It should be possible to extend the construction to boost, but this is work in progress with R. Javadinezhad and U. Kol [18]. We can do the construction quantum mechanically without particular complications, so from now on we will use commutators instead of Poisson brackets.

The existence of an automorphism of the algebra of observables that acts as Lorentz transformations on the radiative variables $N_{AB}$ and leaves supertranslations and boundary gravitons invariant was proven in [19]. The argument given in [19] starts by imposing the desired action of Lorentz translations, parametrized by the vector $\xi$. It is summarized by the following equations

$$
\begin{aligned}
[\tilde{Q}_\xi, N_{zz}] &= i\mathcal{L}_\xi N_{zz}, \\
[\tilde{Q}_\xi, N_{\tilde{z}\tilde{z}}] &= i\mathcal{L}_\xi N_{\tilde{z}\tilde{z}}, \\
[\tilde{Q}_\xi, C] &= 0, \\
[\tilde{Q}_\xi, Q[f]] &= 0.
\end{aligned}
\tag{15}
$$

The "improved Lorentz" defined in (15) commutes with supertranslations by construction. To verify that definition (15) is consistent we must check that the Jacobi identity is satisfied. It is easy to see that the only nontrivial equation to check is

$$
[Q[f], [\tilde{Q}_\xi, C]] + [C, [Q[f], \tilde{Q}_\xi]] + [\tilde{Q}_\xi, [C, Q[f]]] = 0.
\tag{16}
$$

Since $Q[f]$ and $C$ commute to a c-number, eq. (16) is in fact satisfied.

A charge is an operator acting on a Hilbert space, so the construction reviewed above, which proves the existence of an automorphism of the algebra of observables, shows that a charge may exist, but it does not prove that it does. We will show that for rotations such a charge exists by explicitly constructing an angular momentum operator that commutes with the supertranslations $Q[f]$ and the boundary graviton $C$. We will leave the construction of Lorentz boost charges to [18].

We define the angular momentum as in e.g. [12]

$$
Q(Y) = \frac{1}{8\pi G} \int_{I_-^+} d^2\Theta \sqrt{h}\, Y^A(\Theta) N_A,
\tag{17}
$$

with $D_A Y^A = 0$. The derivative $\partial_u N_A$ can be expressed in terms of the independent degrees of freedom using the constraint following from the $uA$ components of the Einstein equations

$$
\begin{aligned}
\partial_u N_A &= -\frac{1}{4} D^B \left( D_B D^C C_{CA} - D_A D^C C_{CB} \right) - u\partial_u \partial_A m_B - T_{uA}, \\
T_{uA} &= -\frac{1}{4}\partial_A \left( C_{BD} N^{BD} \right) + \frac{1}{4} D_B \left( C^{BC} N_{CA} \right) - \frac{1}{2} C_{AB} D_C N^{BC} + 8\pi \lim_{r\to\infty} r^2 T_{uA}^M.
\end{aligned}
\tag{18}
$$

The derivative $\partial_u m_B$ can also expressed in terms of the independent degrees of freedom by use of the $uu$ component of Einstein's equations

$$
\begin{aligned}
\partial_u m_B &= \frac{1}{4} D^A D^B N_{AB} - T_{uu}, \\
T_{uu} &= \frac{1}{8} N_{AB} N^{AB} + 4\pi \lim_{r\to\infty} r^2 T_{uu}^M,
\end{aligned}
\tag{19}
$$

where $T_{\mu\nu}^M$ in (18,19) is the stress-energy tensor of matter. We will assume for simplicity that all asymptotic degrees of freedom of our theory are massless, so that future null infinity $I^+$ is a complete Cauchy surface, all curvature tensors revert to the vacuum at $u \to +\infty$ and $\lim_{u\to+\infty} N_A(u) = 0$. The last property together with $D_A Y^A = 0$ allows the angular momentum to be rewritten as

$$
Q(Y) = \frac{1}{8\pi G} \int_{I^+} du\, d^2\Theta \sqrt{h}\, Y^A(\Theta) \left[ -\frac{1}{4} D^B \left( D_B D^C C_{CA} - D_A D^C C_{CB} \right) - T_{uA} \right].
\tag{20}
$$

This is important for the next step, which leads to a definition of angular momentum satisfying the commutation relations (16) and therefore independent of the arbitrary choice of vacuum made to arrive at eq. (20).

The first property needed to construct the charge is that $N_{AB}(u)$ commutes with the boundary graviton, so $[\int_{-L}^{L} du N_{AB}(u), C] = \int_{-L}^{L} du[N_{AB}(u), C] = 0 \; \forall L.$ [1]. Next, we replace $C_{AB}$ everywhere in eq. (20) with $\check{C}_{AB} \equiv \int_{-\infty}^{u} du' N_{AB}(u')$. The replacement does not affect the term linear in $C_{AB}$ in (20) when the boundary graviton obeys eq. (4), but it does affect the quadratic term present in $T_{uA}$ as per eq. (18). This redefined shear commutes with the boundary graviton $C^{\infty}$ and with $Q_s[f]$ –as $N_{AB}$ does too– but not with the supertranslation $Q[f]$. The redefined charge $\check{Q}(Y)$ commutes with $C^{\infty}$ but has the same commutation relations as $Q(Y)$ with all radiative and matter variables. To get an operator that commutes with $Q[f]$ we "dress" radiative variables and matter fields with the unitary operator $U$. It acts on the radiative variables $N_{AB}$ and matter fields $\phi$ in $T_{\mu\nu}^M$ as [20]

$$UN_{AB}(u,\Theta)U^{-1} = N_{AB}(u - C(\Theta),\Theta), \qquad U\phi(u,\Theta)U^{-1} = \phi(u - C(\Theta),\Theta). \tag{21}$$

The dressing, defined on any operator $O$ by $\hat{O} = UOU^{-1}$ is obviously an automorphism of the operator algebra, so all commutation relations obeyed by undressed operators are satisfied also by dressed ones. So in particular $[\hat{Q}(Y), \hat{N}_{AB}] = i\mathcal{L}_Y N_{AB}$ and the commutator $[\hat{Q}(Y), \hat{Q}(Y')]$ satisfy the algebra of rotations. By replacing everywhere in (20) $C_{AB}$ with $\hat{C}_{AB} = U\check{C}_{AB}U^{-1}$ we finally get a charge that acts correctly on dressed radiative variables and matter fields but also commutes with $C^{\infty}$ and $Q[f]$. Notice that a shift $u \to u - C(\Theta)$ in the variable of integration of (20) does not change the integral so we also have

$$\hat{Q}(Y) = \check{Q}(Y). \tag{22}$$

Even if it is not apparent from the definition given here, this formula agrees with the explicit invariant angular momentum defined in ref. [21] as it will be shown in [18]. An explicit calculation detailed in ref. [18] also shows that the value of $\check{Q}(Y)$ on the vacuum is zero, independently of the value of the soft variables. This property, as well as an invariant definition of boosts, for which many of the simplifications used here do not work, will be studied in the forthcoming paper [18].

## Acknowledgments

This paper reports original work done with U. Kol and R. Javadinezhad, whose collaboration is gratefully acknowledged. M.P. is supported in part by NSF grant PHY-1915219.

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
