# Peer review of "Charges in the Extended BMS Algebra: Definitions and Applications"

_SciPost Physics Proceedings, doi:SciPost Phys. Proc. 4, 005 (2021)_

## Round 1 · Referee Report · Anonymous · 2021-1-14

Report

One of the requested changes (boundary conditions necessary for the existence of topological charges) has been addressed. The request for a physical explanation of the complex gauge transformation in real terms (especially for an observer) has not been addressed. As explained in the first report, the proceedings is clearly written. Since the authors plans on submitting a further paper on the topic, the unaddressed suggestions can be kept for further research. This proceedings summarizes past research by the author.

---

## Round 1 · Author Response

Clarifications after eq. (7) and (13) on the definition and meaning of gauge charges.
The limit due to the new gauge symmetry being defined only asymptotically is made more explicit in the third bullet point after (13)

---

## Round 1 · List of Changes

Clarifications after eq. (7) and (13) on the definition and meaning of gauge charges and observables.

You are currently on this page

Resubmission scipost_202012_00003v1 on 2 December 2020

---

## Editorial Decision

published